# Gut Microbiota-Derived PGF2α Fights against Radiation-Induced Lung Toxicity through the MAPK/NF-κB Pathway

**DOI:** 10.3390/antiox11010065

**Published:** 2021-12-28

**Authors:** Zhi-Yuan Chen, Hui-Wen Xiao, Jia-Li Dong, Yuan Li, Bin Wang, Sai-Jun Fan, Ming Cui

**Affiliations:** 1Tianjin Key Laboratory of Radiation Medicine and Molecular Nuclear Medicine, Institute of Radiation Medicine, Chinese Academy of Medical Sciences and Peking Union Medical College, Tianjin 300110, China; czy.Ph.D@student.pumc.edu.cn (Z.-Y.C.); dongjiali@irm-cams.ac.cn (J.-L.D.); liyuan@irm-cams.ac.cn (Y.L.); wangbinpumc@126.com (B.W.); 2Department of Microbiology, College of Life Sciences, Nankai University, Tianjin 300071, China; xiaohw@nankai.edu.cn

**Keywords:** radiation pneumonia, gut microbiota, gut-lung axis, gut microbiota metabolites, PGF2α

## Abstract

Radiation pneumonia is a common and intractable side effect associated with radiotherapy for chest cancer and involves oxidative stress damage and inflammation, prematurely halting the remedy and reducing the life quality of patients. However, the therapeutic options for the complication have yielded disappointing results in clinical application. Here, we report an effective avenue for fighting against radiation pneumonia. Faecal microbiota transplantation (FMT) reduced radiation pneumonia, scavenged oxidative stress and improved lung function in mouse models. Local chest irradiation shifted the gut bacterial taxonomic proportions, which were preserved by FMT. The level of gut microbiota-derived PGF2α decreased following irradiation but increased after FMT. Experimental mice with PGF2α replenishment, via an oral route, exhibited accumulated PGF2α in faecal pellets, peripheral blood and lung tissues, resulting in the attenuation of inflammatory status of the lung and amelioration of lung respiratory function following local chest irradiation. PGF2α activated the FP/MAPK/NF-κB axis to promote cell proliferation and inhibit apoptosis with radiation challenge; silencing MAPK attenuated the protective effect of PGF2α on radiation-challenged lung cells. Together, our findings pave the way for the clinical treatment of radiotherapy-associated complications and underpin PGF2α as a gut microbiota-produced metabolite.

## 1. Introduction

Currently, lung cancer morbidity and mortality rank first among malignant tumours worldwide [1]. Radiotherapy, as a first-line remedy, is widely used to treat thoracic cancers [2,3]. Although radiotherapy plays a necessary role in the primary and adjuvant treatment of lung cancer, normal lung tissue outside the tumour is sensitive to ionizing radiation and is susceptible to harmful effects, including radiation pneumonia and lung fibrosis [4,5,6] and oxidative stress damage. Approximately 30–40% of lung cancer patients suffer from radiation pneumonitis following radiotherapy, which remains a major challenge for the successful management of lung cancer treatment for local recurrence and distant metastasis [7]. Radiation-induced pneumonitis is characterized by the appearance of a large number of inflammatory cells causing inflammation [8]. After two weeks of radiation treatment, DNA oxidative damage and TGF-β1 (transforming growth factor-β) secretion increase, and other related factors aggravate the inflammatory response [9,10]. Studies have shown that after a single 12 Gy ionizing radiation exposure to the chest of mice, pulmonary TGF-β1 reaches its maximum content in 2 to 4 weeks [11]. Radiation pneumonitis remains a clinical conundrum halting therapy prematurely and reducing the quality of life of patients.

The gut microbiota produces a spectrum of metabolites that permeate into the circulatory system and modulate the physiopathological status of distal organs. For instance, the lungs detain and respond to enteric flora-derived components in peripheral blood, which is known as the gut-lung axis [12]. Lipopolysaccharides, endotoxins from the outer membrane of gram-negative bacteria in the intestine, impact the response of the lung to allergens in asthmatic patients [13]. Short-chain fatty acids (SCFAs, including formic acid and acetic acid, etc.) are the most abundant bacteria-derived metabolites from the fermentation of indigestible fibre-rich diets by specific microorganisms in the caecum and colon. Valeric acid, a kind of SCFAs, increased the survival rate of irradiated mice, protected haematopoietic organs from damage and improved gastrointestinal (GI) function and intestinal epithelial integrity in irradiated mice [14]. As a classic member of the SCFA family, butyrate enters the peripheral circulation, enhances the haematopoietic function of anti-inflammatory macrophage precursors in lung tissues and resists lung influenza virus infection [15]. In addition, the gut-lung axis involves the migration of immune cells from the intestine to the respiratory tract through the circulatory system [16,17]. Our previous studies have identified that gut microbiota-produced metabolites protect against radiation-induced bone marrow toxicity and GI tract toxicity [18]. However, whether the gut microbe-derived metabolites can be employed to mitigate radiation pneumonitis via the gut-lung axis is still not fully understood.

In this study, we aimed to identify the relationship between the gut microbiome and radiation-induced lung injury and screen for relative metabolites with radioprotective functions. Our observations illustrate that faecal microbiota transplantation (FMT) improved lung function, ameliorated radiation pneumonia and scavenging oxidative stress in mouse models. Intriguingly, intestinal flora-produced prostaglandin F2α (PGF2α) inhibited the apoptosis of lung cells by activating MAPK/NF-κB signalling. Altogether, our findings provide novel insight into gut microbiota metabolites and pave the way for fighting against radiation pneumonia in preclinical settings.

## 2. Materials and Methods

### 2.1. Mice

Six to eight-week-old-male (around 20 g) C57BL/6J mice were obtained from Huafukang Bio. (Beijing, China). Mice were maintained in the Specific Pathogen Free (SPF) level animal facility, all experiments were approved by the Institute of Radiation Medicine of the Chinese Academy of Medical Sciences (IRM-CAMS). Mice were housed in standard conditions with ambient temperature 22 ± 2 °C and a 12-h light/dark cycle. The mice had access to a standard food and sterile water ad libitum. Animal experiments were performed according to the institutional guidelines of the Animal Care and Ethics Committee of IRM-PUMC. All animals were randomly divided into three groups (Con, TLI and TLI + FMT). The Con and TLI groups were given 200 μL saline by oral as control. The TLI + FMT group was treated with 200 μL faecal dissolving fluid via oral route (0.1 g faeces dissolved in 2 mL saline).

### 2.2. Cell Culture

Human-derived lung normal epithelial cells (BEAS-2B, CRL-9609, ATCC) and Mouse-derived lung normal epithelial cells (MLE-12, CRL-2110, ATCC) were maintained in 10% foetal bovine serum (Gibco, Grand Island, NY, USA), supplemented with 1% penicillin and 100 mg/mL streptomycin and grown at 5% CO_2_ and 37 °C.

### 2.3. Irradiation Study

All irradiation experiments used a Gammacell-40 137Cs irradiator (Atomic Energy of Canada Limited, Chalk River, ON, Canada) with 0.8 Gy/min. Mice treated with total lung irradiation (TLI) were exposed to a single 15 Gy. Mice were administered intragastrically with PGF2α (Aladdin, D136300, CAS: 38562-01-5, Shanghai, China; around 200 μL per mouse) and the FMT group used the supernatant collected from the Con group’s faeces. It was mixed and dissolved for gavage treatment for 10 days. The lung tissue samples were collected after the 21-day course of the experiment. BEAS-2B and MLE-12 cells experiments were exposed to a single 6 Gy. Control mice and cells received sham irradiation.

### 2.4. RNA Interference

Small interfering RNAs (siRNAs), targeting p38α (Human: siMAPK-1: GCCCAUAAGGCCAGAAACU, siMAPK-2: CAAAUUCUCCGAGGUCUAA and Mouse: siMAPK-1: CGTTCTACCGGCAGGAGCT; siMAPK-2: CATAATTCACAGGGACCTA) and a negative control siRNA (siNC, UUCUCCGAACGUGUCACGU) were obtained from Genewiz (Tianjin, China). Groups of cells were transfected with a mixture of RFect (#11011, Changzhou, China) transfection reagent and siRNA fragments according to the instructions, and samples were extracted after 48 h. The knockdown efficiency of siMAPKs was assayed by protein level (Western blotting analysis).

### 2.5. Western Blotting

Extraction of total protein was made from the lung cells using RIPA (Solarbio, Beijing, China) buffer and centrifuged at 12,000× *g*. Cell protein was separated by 10% SDS-PAGE, and then transferred onto polyvinylidene fluoride (PVDF) membranes. The membranes were blocked with 10% skimmed milk (P0216, Beyotime, Shanghai, China) and incubated with primary antibodies. The monoclonal antibody was obtained from Abcam (Cambridge, MA, USA). HRP conjugated AffiniPure goat anti-rabbit IgG (H + L) was obtained from Proteintech (Proteintech Group, Rosemont, IL, USA). Detailed antibody information is in the Appendix A.

### 2.6. Enzyme-Linked Immunosorbent Assay (ELISA)

The fresh lung tissue samples were weighed and broken up in a tissue crusher, and added to an appropriate amount of PBS to a final concentration of 0.1 g/mL and then centrifuged at 3000 rpm at 4 °C for 15 min. The supernatants were collected and used to measure the protein levels according to the protocol. The ELISA kit manufacturer of TGF-β1 (ml0021115), TNF-α (ml002095), IL-6 (ml002293), IL-1 (ml063132), IL-18 (ml063131) and PGF2α (ml037541) were obtained from MLBio (Shanghai, China). Optical density was read at 450 nm (Rayto, Shenzhen, China).

### 2.7. Quantitative Real-Time Polymerase Chain Reaction (qRT-PCR)

Total RNA was extracted from BEAS-2B cells, MLE-12 cells and lung tissues using TRIzol reagent (Invitrogen, Carlsbad, CA, USA) based on the protocol. In brief, cells and tissues were ground up with TRIzol and then the supernatant was collected by centrifugation with vigorous shaking of trichloromethane, and isopropanol was used to precipitate the RNA. Finally, the RNA precipitate was washed with 75% ethanol and the RNA was lysed in DEPC water. The purity and concentration of the RNA was assessed using a NanoDrop 2000 spectrophotometer (Thermo Scientific, Waltham, MA, USA). The cDNA was synthesized by reverse transcription of the extracted competent RNA using the PrimeScript RT kit (Takara-Bio, Shiga, Japan) according to the instructions, with the following procedure: 42 °C, 30 min, 85 °C, 5 s. qRT-PCR was performed by mixing Fast Start Universal SYBR Green Master (Rox) (Roche Diagnostics GmbH, Mannheim, Germany). The primers used in this study are listed in Appendix A. GAPDH served as the internal reference gene.

### 2.8. Masson Staining and Sirius Red Staining

Freshly cut lung tissue of appropriate size was placed in an embedding box and fixed in 4% paraformaldehyde solution for 24 h and dehydrated using gradient alcohol and xylene after rinsing with distilled water. Then, the dehydrated tissues were embedded in paraffin and cut into 4-µm slices to perform Masson and Sirius red staining. Sequential paraffin sections were dewaxed and stained with Masson or Sirius red according to the instructions provided by the Masson’s Trichrome Stain Kit (Solarbio, G1340, Beijing, China) or Picro Sirius Red solution (G-CLONE, RS1220, Beijing, China), respectively. Images were obtained using a CX31 microscope (Olympus, Tokyo, Japan).

### 2.9. Immunofluorescent Staining

Fresh lung tissue was fixed in 4% paraformaldehyde for 24 h, normally dehydrated and embedded in sections, then the tissue slices were baked at 65 degrees for 60 min, dewaxed and soaked in 4% hydrogen peroxide for 10 min, in 95 degrees citric acid repair solution for 15 min, then closed using goat serum for 1 h, the primary antibody (JNK: ab31419, 1 µg/mL; p38: ab185145, 1/100, abcam, Cambridge, Cambridgeshire, UK) was added dropwise to the tissue slices in proportion, incubated overnight at 4 degrees. Then the corresponding secondary antibody (ab150075, abcam, USA) was added dropwise for 1 h, and finally DAPI staining solution was added dropwise for 15 min, the slices were sealed with anti-burst sealer and photographed by confocal microscopy (LEICA TCS SP8, Weztlar, Germany).

### 2.10. Flow Cytometry Analysis

For cell apoptosis assay, the 1 × 10^4^ lung cells were inoculated in 6-well plates. Then the lung cells were collected in EP tubes, and washed with ice-cold PBS and resuspended with binding buffer (BD Bioscience, Franklin Lakes, NJ, USA). Finally, 2 µL of AnnexinV-FITC (BD Bioscience, Franklin Lakes, NJ, USA) and PI staining solution (BD Bioscience, Franklin Lakes, NJ, USA) were added to each sample and incubated together for 15 min while protected from light, followed by detection using a BD flow cytometer (BD, Franklin Lakes, NJ, USA).

### 2.11. Colony Formation Assays

For lung cell colony formation analysis, 500 lung cells were inoculated in 6-well plates, irradiated/administered with the treatment and then incubated for 2 weeks to observe cell growth. Lung cell colonies were then fixed in 99% methanol for 20 min and stained with Giemsa (Solarbio, Beijing, China) for 1 h.

### 2.12. Cellular Immunofluorescence

BEAS-2B cells and MLE-12 cells were cultured on sterile coverslips in 12-well plates overnight. The cells were fix with 4% paraformaldehyde for 15 min and permeabilized in 0.5% Triton X-100 for 30 min at room temperature. Then, the cells were blocked with normal goat serum (ORIGENE, SP-9000, Beijing, China) for 30 min at room temperature and incubated with anti-NF-κB antibody (SANTA CRUZ, Dallas, TX, USA, #SC-8008) overnight at 4 °C. The cells were washed 3 times with PBS and then incubated with goat anti-rabbit IgG Alexa Fluor^®^ 647 (Abcam, ab150115, Cambridge, UK) in the dark for 1 h at 37 °C. A fluorescent sealing liquid solution, containing DAPI was added to mark nuclei. The images were observed and collected using a fluorescence microscope (Invitrogen EVOS™ FL Auto 2 Imaging System, AMAFD2000, Thermofisher, Waltham, MA, USA).

### 2.13. Measurement of Malondialdehyde

The levels of malondialdehyde (MDA) in lung tissues were assessed using a detection kit from Solarbio (BC0020, Beijing, China) according to the manufacturer’s instructions. The levels of MDA in lung tissues were evaluated and calculated by the following formula, according to the manufacturer’s instructions. Application of MDA was used in the detection of mouse lung tissue.

### 2.14. Respiratory Metabolism

The 24-h RQ (O_2_/CO_2_) containing oxygen intake (VO_2_ (mL/min/kg^0.75) and carbon dioxide emissions (VCO_2_ (mL/min/kg^0.75) was monitored using the Panlab Oxylet system (Panlab, OXYLET, Barcelona, Spain). The Oxylet system was calibrated before assessments as follows: the range of O_2_ was 20% to 50% and the range of CO_2_ was 0% to 1.5%. The housing facility was maintained at a temperature around 22 °C with a 12-h light/12-h dark cycle and continuous access to a standard diet and water. Each mouse was weighed, and the average value was calculated and then housed in the metabolic cage for 24 h.

VO_2_ and VCO_2_ weighted complete equations:$$VO_{2}=\frac{\left(F\times \frac{{\left[O_{2}\right]}_{e}}{100}\right)-\frac{\frac{{\left[O_{2}\right]}_{s}}{100}\times F\times \left(1-\frac{{\left[O_{2}\right]}_{e}}{100}-\frac{{\left[{CO}_{2}\right]}_{e}}{100}\right)}{\left(1-\frac{{\left[O_{2}\right]}_{s}}{100}-\frac{{\left[CO_{2}\right]}_{s}}{100}\right)}}{W^{k}}$$
$$\;\;VCO_{2}=\frac{\frac{F\times \left(1-\frac{{\left[O_{2}\right]}_{e}}{100}-\frac{{\left[CO_{2}\right]}_{e}}{100}\right)}{\left(1-\frac{{\left[O_{2}\right]}_{s}}{100}-\frac{{\left[CO_{2}\right]}_{s}}{100}\right)}\times \frac{{\left[CO_{2}\right]}_{s}}{100}-\frac{{\left[CO_{2}\right]}_{e}}{100}\times F}{W^{k}}$$

RQ weighted equations:$$RQ=\frac{VCO_{2}}{VO_{2}}$$

### 2.15. Bioinformatics Analysis of Gut-Microbiota Sequencing

(1) Bacterial diversity analysis. In vivo experiments contained 12 mice per group, with 6 mice sharing a cage. Three fresh, faecal pellets were collected in each of the two cages to avoid a co-cage effect. Here the control group (Con), the irradiated group (TLI) and the FMT irradiated group (TLI + FMT) were included. Bacterial diversity analysis was conducted as described previously [18]. Briefly, fresh faecal samples were collected and stored at −80 °C. The DNA was extracted using the Power Fecal^®^ DNA Isolation Kit (MoBio Carlsbad, San Diego, CA, USA). Then, the Qiagen Gel Extraction Kit (Qiagen, Germantown, MD, USA) was used to purify the mixture and PCR products in equidensity ratios. The 16S ribosomal RNA (rRNA) V4 gene was sequenced and analysed to assess the gut bacterial diversity using Illumina HiSeq (Novogene Bioinformatics Technology Co., Ltd, Beijing, China). Uparse software (Uparse v7.0.1001, http://drive5.com/uparse/ (accessed on 10 July 2021)) was used to performed sequences analysis. Sequences with more than 97% similarity were assigned to the same OTUs. Representative sequence for each OTU was screened for further annotation. The Silva123 Database was used based on the RDP classifier (Version 2.2, http://sourceforge.net/projects/rdpclassifier/ (accessed on 10 July 2021)) algorithm to annotate taxonomic information for each representative sequence. The primers used in bacterial diversity analysis are listed in Appendix A.

(2) Untargeted Metabolomics. Fresh faeces were quick-frozen with liquid nitrogen and then suspended with prechilled 80% methanol and 0.1% formic acid by vortexing. The samples were incubated on ice for 5 min and centrifuged for 5 min (15,000 rpm, 4 °C). The supernatant was collected and diluted by LC-MS grade water to a concentration containing 60% methanol and transferred into a fresh tube with 0.22 µm filter. Then, the samples were centrifuged for 10 min (15,000× *g*, 4 °C). LC-MS/MS analyses were performed using a Vanquish UHPLC system (Thermo Fisher) coupled with an Orbitrap Q Exactive HF-X mass spectrometer (Thermo Fisher). The filtrate was injected into an Hyperil Gold column (100 × 2.1 mm, 1.9 μm) using a 16-min linear gradient at a flow rate of 0.2 mL/min. The eluents for the positive polarity mode were eluent A (0.1% FA in Water) and eluent B (methanol), and those for the negative polarity mode were eluent A (5 mM ammonium acetate, pH 9.0) and eluent B (methanol). The solvent gradient was set as follows: 2% B, 1.5 min; 2–100% B, 12.0 min; 100% B, 14.0 min; 100–2% B, 14.1 min; 2% B, 16 min. Q Exactive HF-X mass spectrometer was operated in positive/negative polarity mode with spray voltage of 3.2 kV, capillary temperature of 320 °C, sheath gas flow rate of 35 arb and aux gas flow rate of 10 arb. The raw data were generated by UHPLC-MS/MS using the Compound Discoverer 3.0 (CD 3.0, Thermo Fisher) to perform peak alignment, peak picking and quantitation for each metabolite. The main parameters were set as follows: retention time tolerance, 0.2 min; actual mass tolerance, 5 ppm; signal intensity tolerance, 30%; signal/noise ratio, 3; and minimum intensity, 100,000. Peak intensities were normalized to the total spectral intensity. The normalized data were used to predict the molecular formula on the basis of additive ions, molecular ion peaks and fragment ions. Then, peaks were matched with the mzCloud (https://www.mzcloud.org/ (accessed on 6 June 2021)) and ChemSpider (http://www.chemspider.com/ (accessed on 6 June 2021)) databases to obtain accurate qualitative and relative quantitative results.

### 2.16. Statistical Analysis

Results are expressed as mean ± SD from three independent experiments, and data significance between independent groups was analysed by Student’s *t*-test and Wilcoxon rank sum test using SPSS 27.0 software (Chicago, IL, USA), ** p* < 0.05, *** p* < 0.01 and **** p* < 0.001 representing statistically significant differences in data.

## 3. Results

### 3.1. Faecal Microbiota Transplantation Fights against Radiation Induced Lung Toxicity

Wild-type C57BL/6J mice were exposed to a single 15 Gy dose γ-ray local chest irradiation to mimic radiotherapy for lung cancer, and faecal microbiota transplantation (FMT) was performed on the next day for 10 days, based on our previous study [19]. As shown in Figure 1A, FMT prevented the weight loss induced by total lung irradiation (TLI), implying that FMT might improve the prognosis of radiotherapy. Lung tissue staining (Sirius Red and Masson staining) showed alveolar destruction and fibrin accumulation in irradiated mice, while FMT attenuated structural lung damage (Figure 1B). FMT erased the lung coefficient elevated by local chest irradiation (Figure 1C), suggesting that FMT may improve pulmonary respiratory function after radiotherapy. In fact, the respiratory quotient (RQ) value increased after FMT and decreased the VO_2_ intake but did not change VCO_2_ emissions following the radiation challenge (Figure 1D,E and Appendix A). Importantly, FMT alleviated radiation pneumonia, representing a reduction in the inflammatory status (Figure 1F,G) and scavenging of oxidative stress (MDA and SOD) (Figure 1H and Appendix A). Altogether, our observations demonstrate that FMT improves lung respiratory function and relieves pneumonia following local chest irradiation.

### 3.2. FMT Shapes the Gut Microbiota Configuration of Mice after Local Chest Irradiation

To elucidate the underlying mechanisms by which FMT battles against radiation-induced lung toxicity, 16S rRNA sequencing was performed to analyse the effects of FMT on the intestinal bacterial community following radiation stimuli. Intriguingly, local chest irradiation heightened the observed species of microbes in faecal pellets, while FMT erased the elevation (Figure 2A). The Chao1, ACE and Shannon indices further validated the results (Figure 2B–D). Specifically, the irradiated mice showed an enrichment in *Akkermansia, Desulfovibrio, Parasutterella* and a decline in *Rikenella* at the genus level; however, FMT reversed these alterations (Figure 2E). The weighted UniFrac analysis revealed that local chest irradiation increased the β-diversity of gut microbiota in the experimental mice with or without FMT (Figure 3A). In the unweighted UniFrac analysis, radiation exposure slightly decreased the β-diversity, which was attenuated by FMT (Figure 3B). Principal component analysis (PCA) and unweighted/weighted principal coordinate analysis (PCoA) further showed separations among the three cohorts (Figure 3C–E). All evidence suggests that local chest irradiation restructured the gut microbiota configuration in the mouse models. Specifically, the irradiated mice harboured a lower abundant of *g_Parabacteroides_s_ Parabacteroides_distasonis*, but a higher abundance of *g_Parabacteroides_s_Parabacteroides_goldsteinii* and *g_Faecalibaculum_s_ Faecalibaculum_rodentium* in the GI tract at the species level (Figure 3F–H).

### 3.3. FMT Remoulds the Gut Microbiota Metabolome Fluctuated by Local Chest Irradiation

Next, we further analysed the changes in the gut microbial metabolome using LC-MS/MS. A volcano plot showed that local chest irradiation disordered the metabolite profile of gut microorganisms, and as expected, FMT educated the intestinal microbial taxonomic proportions and tuned the metabolite profile (Figure 4A,B). A heatmap was used to further describe the details of the metabolome changes. For example, total lung irradiation increased the relative abundance of calophyllolide and methyl aklanonate and decreased that of primaquine and tridemorph (Figure 4C). FMT elevated the frequency of arnebinol and piperidine and reduced that of 3-lodobenzoic acid and prolinebutyl ester (Figure 4D). All the results further validated that local chest irradiation indeed shifted the gut microbiota community. Given that gut microbe-derived metabolites are significant signalling messengers linking the gut to distal organs, we hypothesized that FMT-mitigated radiation-induced lung injury might depend on gut microbiota metabolites. Then, we screened for potential metabolites based on the strategy that the gut microbiota metabolites should be decreased with local chest irradiation alone and increased following FMT. In light of this principle, four target metabolites were obtained and are displayed in Figure 4E. We preliminarily assessed the radioprotective effects of these metabolites. The clone formation assays showed that PGF2α protected cells against radiation exposure (Figure 4F). Given prostaglandin-endoperoxide synthase 2 (PTGS2) is a key modulator of PGF2α synthesis, the bioinformatics analysis showed that lung cancer tissues carried a lower expression of PTGS2 than peritumoural tissues (Figure 4G,H; https://ccsm.uth.edu/miRactDB (accessed on 16 November 2020)). In addition, lung cancer patients with a high expression of PTGS2 had a higher overall survival rate (Figure 4I,J; http://gepia.cancer-pku.cn/index.html (accessed on 16 November 2020)), implying that high levels of PTGS2 and downstream PGF2α predict a good prognosis. Based on the above evidence, we focused on PGF2α in our further studies.

### 3.4. The Intestinal Flora Metabolite PGF2α Improves Radiation-Induced Lung Toxicity

The experimental design is shown in Figure 5A. First, we assessed the levels of PGF2α in the faeces, serum and lung. The results revealed that PGF2α accumulated in the three samples after oral gavage (Figure 5B and Appendix A), implying that PGF2α might play a role in lung tissues. Intriguingly, the PGF2α administration via the oral route lessened body weight loss of local chest irradiated mice (Figure 5C,D). Sirius red and Masson-stained sections of lung tissue showed that the PGF2α treatment prevented structural damage and collagen accumulation in alveoli following the radiation challenge (Figure 5E,F). In parallel with FMT, oral gavage of PGF2α alone reduced the lung coefficient, increased the RQ value, and decreased the VO_2_ intake but did not change the VCO_2_ emissions following total lung irradiation (Figure 5G–I and Appendix A), suggesting that PGF2α might be a potential substitute for FMT to improve lung function after radiotherapy. Finally, we examined the inflammatory status in lung tissues and observed that the PGF2α treatment downregulated the levels of proinflammatory factors (Figure 5J–L and Appendix A). Altogether, our findings indicate that PGF2α might be a radioprotective agent for ameliorating lung respiratory function and fighting against radiation pneumonia.

### 3.5. PGF2α Activates FP/MAPK/NF-κB Signalling Pathway to Inhibit Radiation-Induced Lung Cell Apoptosis

Next, we further explored the underlying mechanism by which PGF2α mitigated radiation induced injury using human bronchial epithelial cells (BEAS-2B). The clone formation assays showed that PGF2α facilitated the proliferation of BEAS-2B cells in a dose-dependent manner after radiation exposure (Figure 6A). Accordingly, 4 μM PGF2α was used as the optimal concentration in the subsequent experiments. A decreased expression of inflammatory factors (TGF-β1, IL-1 and TNF-α) was observed in the PGF2α group (Appendix A–C). The flow cytometry analysis revealed that the radiation challenge increased the number of apoptotic BEAS-2B (from 4.8% to 11.4%); however, the PGF2α addition decreased the number of apototic BEAS-2B (from 11.4% to 4.6%, Figure 6B), suggesting that PGF2α might be able to inhibit radiation-induced apoptosis. PGF2α downregulated the protein level of Caspase-6, a driver of apoptosis, further validating the antiapoptotic roles of PGF2α (Figure 6C). PI3K/AKT and MAPK signalling are involved in apoptosis. Thus, we assessed whether PGF2α inhibited apoptosis through these two pathways. The q-PCR assays showed that PGF2α slightly increased the expression of the F-prostanoid (FP) receptor (Appendix A), downregulated the expression of PI3K and AKT (Figure 6D,E), but upregulated the expression of MAPK, ERK and their downstream gene NF-κB after radiation exposure (Figure 6G–I). The Western blot and immunofluorescent staining assays further validated the results, indicating that PGF2α blocks radiation-elicited apoptosis via MAPK (JNK, p38 and ERK) signalling, but not the PI3K/AKT pathway (Figure 6F,J–L). Given that NF-κB is a transcription factor that needs to directly bind special regions of DNA to modulate gene expression, we examined the location of NF-κB in irradiated BEAS-2B with or without the PGF2α treatment. The immunofluorescence assays showed that PGF2α addition promoted the accumulation of NF-κB in the nucleus (Figure 6M). Together, our observations demonstrate that PGF2α inhibits apoptosis in lung cells to fight against irradiation by activating the FP/MAPK/NF-κB axis.

### 3.6. Blocking MAPK Attenuates the Protective Effect of PGF2α on Lung Cells Following Radiation

To further validate the key role of MAPK/NF-κB signalling in PGF2α mediated radioprotection, siRNAs targeting MAPK and a negative control were transfected into BEAS-2B cells and mouse lung epithelial cells (MLE-12). Figure 7A,F show the transfection efficiency. Given the interference efficacy, siMAPK-2 (BEAS-2B) and siMAPK-1 (MLE-12) were used in the subsequent experiments. As expected, silencing MAPK elevated the number of apoptotic cells and upregulated the expression of apoptotic markers in the irradiated BEAS-2B cells (Figure 7B,C) and MLE-12 cells (Figure 7G and Appendix A) treated with PGF2α. Next, we assessed the expression of downstream MAPK genes. The qRT–PCR and Western blot assays revealed that although PGF2α increased the expression level of MAPK/NF-κB signaling, the MAPK knockdown reduced the levels of JNK, p38, ERK and NF-κB in the irradiated BEAS-2B cells (Figure 7D and Appendix A) and MLE-12 cells (Figure 7H and Appendix A), as well as inhibited the accumulation of NF-κB in the nucleus (Figure 7E). Finally, we analysed the overall survival rate of lung cancer patients based on the expression of MAPK and found that patients carried high levels of MAPK (http://dna00.bio.kyutech.ac.jp/PrognoScan/index.html (accessed on 6 November 2021)) which represented a higher overall survival rate (Figure 7J,K). Altogether, our observations validate that the FP/MAPK/NF-κB axis plays a pivotal role in the radioprotection of PGF2α in lung cells.

## 4. Discussion

Radiation-induced lung toxicity, such as pneumonia and pulmonary fibrosis, is the most common and intractable complication of radiation therapy for patients with chest tumours (such as lung cancer and breast cancer) [20]. To date, no safe and effective remedy is applicable for radiation-induced lung toxicity in clinical scenarios. Gut microbiota-produced bioactive metabolites permeate into the bloodstream and exert pharmacological effects in distant organ systems, such as lung tissue, establishing the gut-lung axis [12,21], and the critical role of the gut-lung axis is well studied [22,23,24]. Although Nie and colleagues reported that FMT could reduce radiation lung injury [25], these authors did not analyse whether FMT restructured the gut bacterial configuration in their study. Thus, which kind of gut microbiome plays a key role in this event and the underlying mechanism remains unclear. Furthermore, these authors only examined the structural changes in lung tissues and missed alterations in the respiratory function, inflammatory status and oxidative stress. Epidemiological investigations show that the incidence of pneumonia is 7.7%, which is much higher than that of pulmonary embolism (1.3%) and chronic obstructive pulmonary disease (1.3%) [26]. The gut microbiota, emerging as the “second genome” and “forgotten organ”, plays important roles in the physiopathological process of the lung [17]. Antibiotic intake and microbiota transplantation are the most effective ways to remould the gut microbiome, which induces profound effects on hosts. Therefore, the analysis of the gut microbiota, following FMT, paves the way for untangling the underlying mechanism by which the gut microbiome battles against radiation-induced lung toxicity. In this study, we focused on the treatment for acute radiation injury. Radiation pneumonia usually occurs within the hours, days and weeks after exposure [27], radiation-induced pulmonary fibrosis can be observed after 4 weeks following radiation exposure [28]. Thus, we assessed the structure, function and inflammation of lung tissues in experimental mice at 21 days after irradiation. Notably, we observed that local chest irradiation shaped the α- and β-diversity of intestinal bacteria, which bolstered the function of the gut-lung axis. We also assessed the enteric bacterial taxonomic proportions after FMT and found that FMT preserved the microflora configuration shifted by total lung irradiation and mitigated radiation-induced lung toxicity. The evidence indicates the following two points: (1) FMT indeed restructured the gut microbiota community; and (2) The modulation of the gut flora impacted the inflammatory status and oxidative stress of lung tissues. Importantly, the adverse side effects associated with radiotherapy not only include inflammation and ROS elevation but also represent a loss of lung function, which reduces the quality of life of patients. Therefore, we also paid attention to the lung respiratory function of experimental mice in this study and found that the RQ value of the irradiated mice was significantly increased after FMT nearly to the normal index, representing improved lung ventilation function.

FMT treatment is the process of transferring the microbial ecology of healthy donors to patients to treat microbial diseases [29], but it is not commonly used as a first-line treatment [30]. Studies have reported that the incidence of adverse events after FMT is 28.5%, of which the upper gastrointestinal tract is 43.6% and the lower gastrointestinal tract is 17.7% [31]. Other disadvantages, including occupying a large number of medical resources, not being suitable for the treatment of a large-scale population, and complicated treatment processes, limit its clinical application [32]. Thus, finding a safe, effective, reliable and convenient-to-operate treatment is an urgent need. The gut microbiota is the source of a series of bioactive metabolites, such as short-chain fatty acids (SCFAs), structuring the gut-brain axis or gut-lung axis to regulate the physiology of distal organs [33]. Given the abovementioned data, we further analysed the metabolome of intestinal microorganisms in this system and screened four target metabolites, namely, micronomicin, prostaglandin f2α, trimethylamine N-oxide and histidine hydrochloride hydrate. Among the four metabolites, PGF2α exhibited the most overt increase in protection in normal lung cells. In addition, micronomicin is an antibiotic [34] mainly used for infections of sensitive gram-negative bacteria and drug-resistant Staphylococcus aureus. TMAO modulates protein activity and stability, increases foam cell production and inhibits reverse cholesterol transport. However, TMAO is a risk factor for cardiovascular disease [35]. Histidine is a nonessential amino acid in adults but an essential amino acid in young children [36]. Imidazole propionate and histamine are acknowledged secondary metabolites of histamine, and histamine is considered a mediator of acute and chronic inflammatory responses. Thus, we focused on PGF2α and found that irradiated mice with PGF2α administration via the oral route improved lung ventilation function and alveoli integrity. Lung cancer patients with a high expression level of PTGS2, a classic PGF2α-related gene, in tumours exhibited a high overall survival rate, implying that PGF2α might be positively correlated with good prognosis in lung cancer patients. All the findings underpin that PGF2α might be employed as a safe and effective succedaneum for FMT to protect patients against radiation-induced toxicity when suitable donors and medical resources are unavailable in preclinical settings.

PGF2α and PI3K are competitors that bind FP receptors. Therefore, an increase in cell surface receptors with more PGF binding indicates a decrease in PI3K coalescing and inhibition of relative signalling [37]. In this study, irradiation activated the PI3K/AKT signalling pathway. However, PGF2α had a deep affinity for the FP receptor, and PI3K/AKT signalling was blocked following the PGF2α addition. The activation of the PI3K/AKT pathway induces a malignant cell phenotype, thereby promoting cell proliferation, invasion, metastasis, angiogenesis and treatment failure [38]. The PIK3/AKT pathway plays a key role in fibrosis activation [39]. Studies have reported that PGF2α can activate MAPK signals to promote the proliferation of endometrial cancer cells by binding the FP receptor [40]. Although the MAPK superfamily mediates various signal transduction pathways stimulated by ionizing radiation, including extracellular signal-regulated kinase (ERK), c-Jun N-terminal kinase (JNK) and p38/MAPK [41], the activation of MAPK/ERK signalling is regarded as an effective regulator of cell growth, differentiation and development [42]. MAPK activation can regulate the production of vascular endothelial growth factor and promote angiogenesis. Several studies have demonstrated that the activation of the MAPK pathway is a central feature of the radioresistance mechanism [41]. Ghrelin therapy increased ERK activation and suppressed JNK activation in ileum, and decreased intestinal injury after radiation injury combined wound (CI) [43]; in brain haemorrhage induced by CI, Ghrelin therapy with pegylated G-CSF significantly mitigated ERK1/2 and JNK activation [44]. In addition, treatment with JNK inhibitor promoted apoptosis in MCF-7 cells exposure to X-ray radiation [45]. Changes in p38/JNK/ERK protein expression following irradiation in different cell lines in different organs have different biological effects on production. NF-κB, a transcription factor, is the downstream gene of the MAPK/ERK pathway. ERK1/2 phosphorylation propels NF-κB to enter the cell nucleus and activate the expression of antiapoptotic proteins [46,47]. In parallel, we found that PGF2α addition upregulated the protein levels of p38, JNK and ERK in normal lung cells and tissues. Cell nuclei differ in size at different stages of the cell cycle. It has been reported that the nuclear volume and the number of nuclear pore complexes (NPCs) are almost double during interphase in dividing cells [48]. We found that NF-κB was located in the cytoplasm with or without radiation stimuli. However, NF-κB was overexpressed and accumulated in the cell nucleus following the PGF2α treatment. This finding does not seem to be related to the size of the nucleus. We used siRNA to inhibit the expression of MAPK and found that PGF2α failed to protect radiation-induced lung cells following the transfection. The cell lines used here were lung epithelial cells and future studies will need to include a variety of cell lines such as lung fibroblasts to reach a more comprehensive conclusion. In addition, the bioinformatics analysis revealed a positive correlation between MAPK, a potential target gene of PGF2α, and the survival rate of lung cancer patients. Patients with high MAPK expression in lung cancer tissues had a higher survival rate, which underpinned our experimental results. Therefore, MAPK/ERK1 may play both a radioprotective role and prolong the survival rate of lung cancer patients.

## 5. Conclusions

PGF2α plays radioprotective roles, such as promoting cell proliferation and inhibiting apoptosis, which might be dependent on the FP/MAPK/NF-κB axis, at least partly. Altogether, our study provides novel insight into gut microbiota-derived metabolites and suggests that gut flora-produced PGF2α might be employed as a potential radioprotective agent to fight against radiation-induced toxicity in preclinical settings.

## Figures and Tables

**Figure 1 antioxidants-11-00065-f001:**
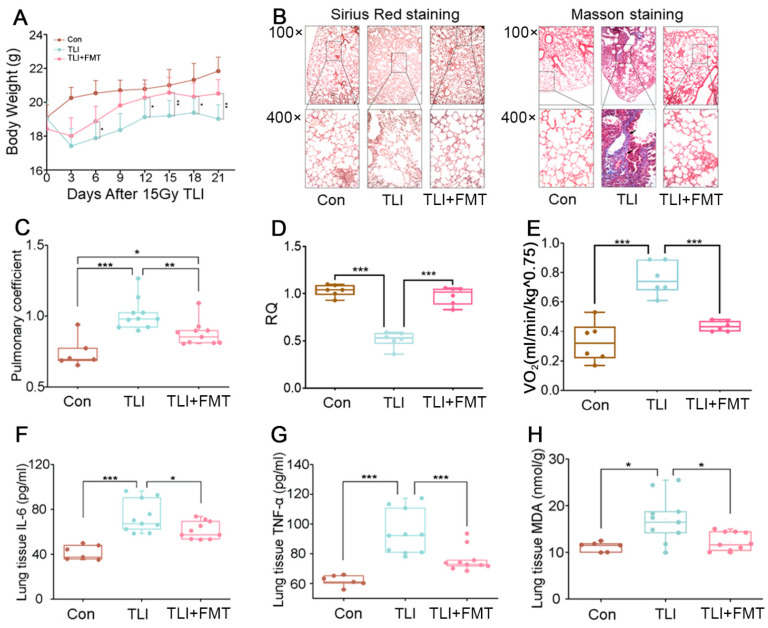
Faecal microbiota transplantation fights against radiation pneumonia in mice: (**A**) The changes in body weight of experimental mice after 15 Gy TLI. (**B**) The lung tissues were stained with Sirius red and Masson (100× and 400×). (**C**) Mouse pulmonary coefficient. (Con: *n* = 6, TLI and FMT: *n* = 10). (**D**,**E**) Respiratory quotient (RQ) and VO_2_ intake of mice in 24 h (*n* = 6). (**F**,**G**) The inflammatory factor of IL-6 and TNF-α in lung tissues from the experimental mice analysed by ELISA (Con: *n* = 6, TLI and FMT: *n* = 10). (**H**) The MDA level in lung tissues of each group was examined (Con: *n* = 6, TLI and FMT: *n* = 10). (* *p* < 0.05, ** *p* < 0.01 and *** *p* < 0.001; Student’s *t*-test; FMT: 200 μL Faecal dissolving fluid/mouse).

**Figure 2 antioxidants-11-00065-f002:**
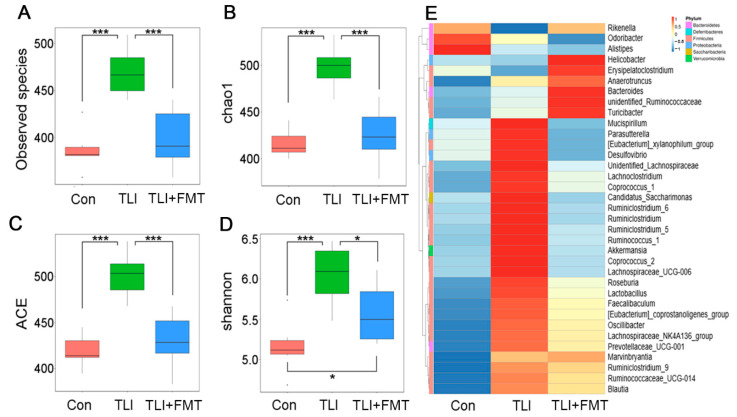
FMT shapes the gut microbiota configuration of mice after local chest irradiation: (**A**–**D**) Analysis of α-diversity by 16S rRNA sequencing of the gut bacteria, mainly the observed species number (**A**), Chao1 diversity index (**B**), ACE diversity index (**C**) and Shannon diversity index (**D**). (**E**) The heat map is colour-based on row Z-scores. The mice with the highest and lowest bacterial level are in red and blue, respectively. (Statistically significant differences are indicated: Wilcoxon rank sum test, *n* = 6 per group, * *p* < 0.05 and *** *p* < 0.001).

**Figure 3 antioxidants-11-00065-f003:**
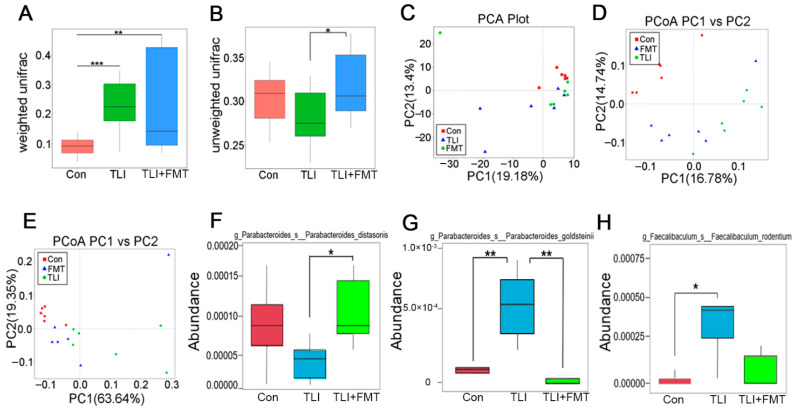
FMT shifts the gut microbiota structure of local chest irradiation mice: (**A**,**B**) The β diversity of enteric bacteria was compared by weighted (**A**) and unweighted (**B**) unifrac analysis. (**C**–**E**) PCoA were used to examine the alteration of intestinal bacteria taxonomic pattern; (**F**–**H**) The relative abundances of *g*_*Parabacteroides_s_ Parabacteroides_distasonis*, *g*_*Parabacteroides_s_Parabacteroides_goldsteinii* and *g_Faecalibaculum_s_ Faecalibaculum_rodentium* at the species level was assessed using 16S high-throughput sequencing after TLI and FMT. (Statistically significant differences are indicated: Wilcoxon rank sum test, *n* = 6 per group, * *p* < 0.05, ** *p* < 0.01 and *** *p* < 0.001).

**Figure 4 antioxidants-11-00065-f004:**
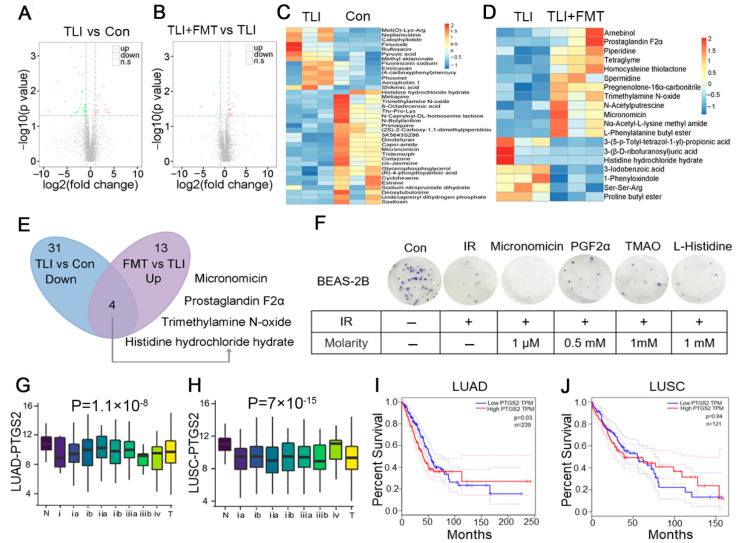
FMT remoulds the gut microbiota metabolome fluctuated by local chest irradiation: (**A**,**B**) Volcano plots of identified different metabolites in the faecal pellets from mice. In the volcano plots, each point represented a metabolite; (**C**) Heatmap of the differences in metabolites of faecal pellets from mice in TLI and Con group; (**D**) Heatmap of the differences in metabolites of faecal pellets from mice in TLI group and FMT group; (**E**) Screening out the metabolites in specific paradigms; (**F**) The effects of PGF2α, Micronomicin, TMAO and L-Histidine on irradiated BEAS-2B cells were measured by clone formation assay; (**G**,**H**) The expression level of PTGS2 in tumour and peritumour was analysed; (**I**,**J**) Kaplan-Meier analysis of the overall survival rate of LUAD and LUSC patients with different expression of PTGS2. *p* < 0.05 by log-rank test between the patients with high and low expression of PTGS2 in LUAD.

**Figure 5 antioxidants-11-00065-f005:**
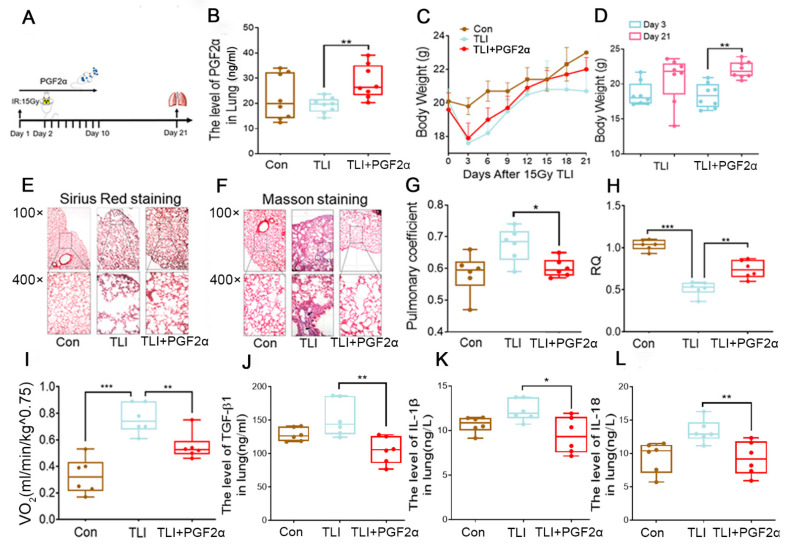
The intestinal flora metabolite PGF2α improves the radiation-induced lung toxicity in mice: (**A**) The experimental roadmap; (**B**) The level of PGF2α in lung pages of mice was analysed by ELISA (*n* = 8); (**C**) Body weight of mice were compared among Con, TLI and PGF2α treatment groups; (**D**) The body weight of mice on the 3rd and 21st day after 15 Gy irradiation with or without PGF2α treatment (*n* = 8); (**E**,**F**) Tissue morphology was observed at ×100 and ×400 using tissue staining (Sirius Red and Masson staining); (**G**) The pulmonary coefficient in three groups of mice; (**H**,**I**) Respiratory quotient (RQ) and VO_2_ intake of mice in each group in 24 h (*n* = 6); (**J**–**L**) The levels of TGF-β1 (**J**), IL-1β (**K**) and IL-18 (**L**) in the lung were measured by ELISA. (* *p* < 0.05, ** *p* < 0.01 and *** *p* < 0.001; Student’s *t*-test).

**Figure 6 antioxidants-11-00065-f006:**
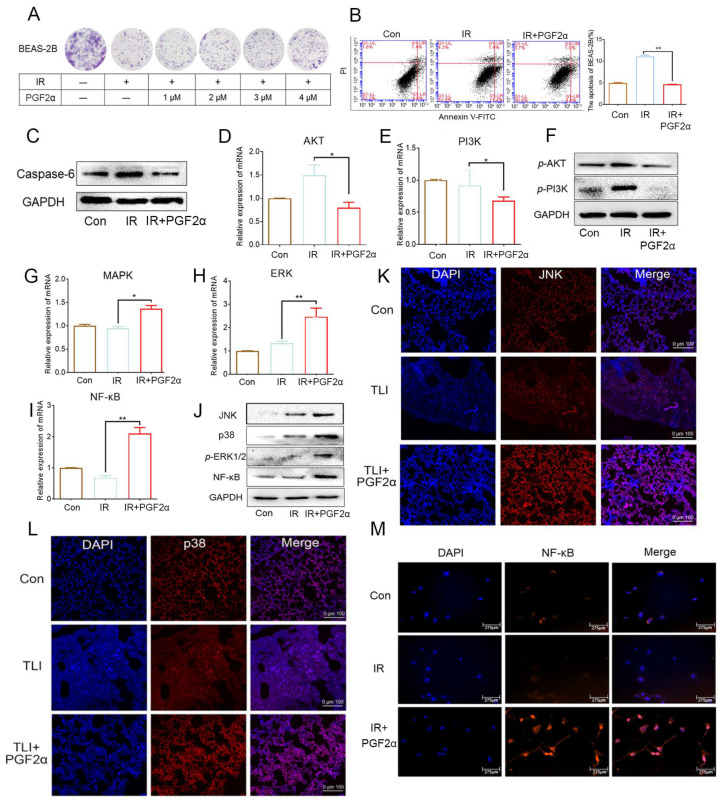
PGF2α activates FP/MAPK/NF-κB signalling pathway to inhibit radiation-induced lung cell apoptosis: (**A**) The effect of PGF2α on radiation-exposed BEAS-2B cells was assessed by clone formation assay. (**B**) The apoptosis of irradiated BEAS-2B cells was analysed by flow cytometry. (**C**) The Caspase-6 expression was examined by Western blotting in irradiated BEAS-2B cells. (**D**–**F**) The expression levels of PI3K and AKT in irradiated BEAS-2B cells with or without PGF2α treatment were measured by qRT-PCR and Western blotting. (**G**–**J**) The expression levels of MAPK, ERK and NF-κB in irradiated BEAS-2B cells with or without PGF2α treatment were measured by qRT-PCR and Western blotting. (**K**,**L**) Protein-level expression of JNK (**K**) and p38 (**L**) of lung tissue (in vivo) were measure by immunofluorescent staining. (**M**) Immunofluorescence showed the expression and location of NF-κB in irradiated BEAS-2B cells. (* *p* < 0.05, ** *p* < 0.01; Student’s *t*-test).

**Figure 7 antioxidants-11-00065-f007:**
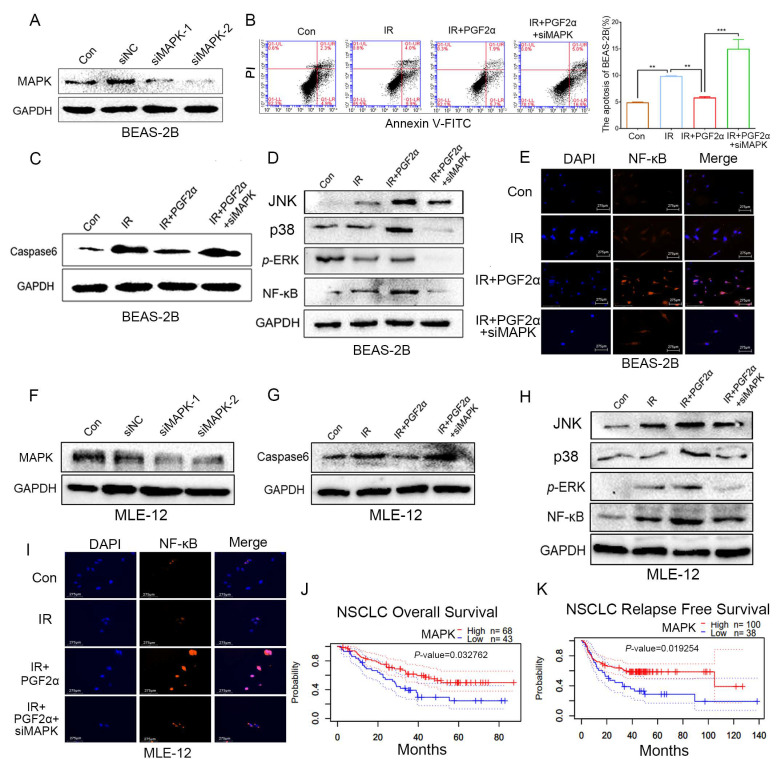
Blocking MAPK attenuates the protective effect of PGF2α on lung cells following radiation: (**A**) The interference efficiency of siRNA targeting MAPK was examined by Western blotting in BEAS-2B cells. (**B**) The apoptosis of irradiated BEAS-2B cells was analysed by flow cytometry. (**C**) The caspase-6 expression was examined by Western blotting in irradiated BEAS-2B cells. (**D**) The expression levels of JNK, p38, *p*-ERK and NF-κB in irradiated BEAS-2B cells were assessed by Western blotting. (**E**) Immunofluorescence showed the expression and location of NF-κB in irradiated BEAS-2B cells. (**F**) The interference efficiency of siRNA targeting MAPK was examined by Western blotting in MLE-12 cells. (**G**) The caspase-6 expression was examined by Western blotting in irradiated MLE-12 cells. (**H**) The JNK, p38, *p*-ERK and NF-κB expression in irradiated MLE-12 cells were examined by Western blotting. (**I**) Immunofluorescence showed the expression and location of NF-κB in irradiated MLE-12 cells. (**J**,**K**) Kaplan–Meier analysis of overall survival and relapse free survival rate of MAPK expression in lung cancer patients. (** *p* < 0.01 and *** *p* < 0.001; Student’s *t*-test).

## Data Availability

Data are contained within the article and Appendix A.

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
