# Peer review of "Gut Microbiota-Derived PGF2α Fights against Radiation-Induced Lung Toxicity through the MAPK/NF-κB Pathway"

_antioxidants, 2021, doi:10.3390/antiox11010065_

Round 1
Reviewer 1 Report
Please highlight the revised areas for this reviewer to review the revised version.
1. Line 68: spell out FMT, because it appears the first time in the introduction section.
2. Line 86: What are BEAS-2B cells and MLE-12 cells? Please include the explanations.
3. Line 351: Figure 5K and L display TGF-ß1 and IL mRNA. There are no data on protein levels., because their protein levels but not mRNA expression are essential. Please measure TGF-ß1, IL-1ß, and IL-18 protein levels and include them in this manuscript.
4. Lines 376-417: The claim is not convincing yet. MAPK contains ERK1/2, JNK, and p38. There are no data on JNK and p38. the authors cannot exclude JNK and p38 unless their data show no changes in JNK and p38 using Western blotting or immunofluorescent staining. So, please include JNK and p38 data in vivo and in vitro.
5. Lines 490-520: This long paragraph sounds reluctant and weak. It is evident that in other organs, JNK and p38 are activated by irradiation (Kiang et al., 2020, PMID: 32426105 and Kiang et al., 2019, PMID: 34368440 ). Therefore, in this discussion section, a comparison between lung and other organs in response to irradiation should be made and included. Your new data on JNK and p38 in the presence of PGF2α should be updated here and indicating the uniqueness or similarity of lung responding to irradiation while compared to other organs.
Reviewer 2 Report
An interesting study and mostly well done.
The authors refer to the treatment modulating pneumontis. If that is the case why to there stain tissues for collagen, a marker of fibrosis. Markers of pneumonitis would include edema and permeability increases, recruitment of inflammatory cells. These should have been included along with the cytokine measures.
Also it is not clear why the 21 day time point was chosed, typically in the C57/Bl6 pneumonitis peaks at 8-12 weeks
The in vitro experiements provide some clarity of the role of PGF2 , but the limited focus on the epithelium does not account for the effects of inflammatory cells of lung fibroblasts.
What was the rationale for using both BEAS-2B and MLE-12 cells?
The discussion regarding PGF2a production by lung tumors seem superfluous as the focus was normal tissue toxicity.
Treatment with FMT or supplemental PGF2 was for 10 days. Why was this time of treatment chosen?
Round 2
Reviewer 1 Report
The revised version has made a significant improvement. Some comments for clarifying the text contents are listed below.
- Line 3: Typo error. Please correct it. Replace “though” with “through”.
- Line 83: Replace “used for” with “randomly divided into three”.
- Line 110: Insert “was made” before “from”.
- Line 119: Delete “,” before “were”.
- Line 140: Replace “is” with “was”.
- Line 207: Replace “are” with “were”.
- Line 347: Insert “the” before ”radiation-induced”.
- Line 364: Insert “the” before “radiation-induced”.
